# Exposure to Zearalenone Leads to Metabolic Disruption and Changes in Circulating Adipokines Concentrations in Pigs

**DOI:** 10.3390/toxins13110790

**Published:** 2021-11-08

**Authors:** Veronika Nagl, Bertrand Grenier, Philippe Pinton, Ursula Ruczizka, Maximiliane Dippel, Moritz Bünger, Isabelle P. Oswald, Laura Soler

**Affiliations:** 1BIOMIN Research Center, BIOMIN Holding GmbH, Technopark 1, 3430 Tulln, Austria; veronika.nagl@dsm.com (V.N.); bertrand.grenier@dsm.com (B.G.); 2Toxalim (Research Centre in Food Toxicology), INRAE, ENVT, INP-Purpan, University of Toulouse, UPS, 31027 Toulouse, France; Philippe.Pinton@inrae.fr (P.P.); Isabelle.Oswald@inrae.fr (I.P.O.); 3University Clinic for Swine, Department for Farm Animals and Veterinary Public Health, University of Veterinary Medicine Vienna, Veterinärplatz 1, 1210 Vienna, Austria; ursula.ruczizka@vetmeduni.ac.at (U.R.); maximiliane.dippel@vetmeduni.ac.at (M.D.); moritz.buenger@vetmeduni.ac.at (M.B.)

**Keywords:** mycotoxins, zearalenone, adipokines, metabolism, estrogen

## Abstract

Zearalenone (ZEN) is a mycotoxin classified as an endocrine disruptor. Many endocrine disruptors are also metabolic disruptors able to modulate energy balance and inflammatory processes in a process often involving a family of protein hormones known as adipokines. The aim of our study was to elucidate the role of ZEN as metabolic disruptor in pigs by investigating the changes in energy balance and adipokines levels in response to different treatment diets. To this end, weaned piglets (*n* = 10/group) were exposed to either basal feed or feed contaminated with 680 and 1620 µg/kg ZEN for 28 days. Serum samples collected at days 7 and 21 were subjected to biochemistry analysis, followed by determination of adipokine levels using a combined approach of protein array and ELISA. Results indicate that ZEN has an impact on lipid and glucose metabolism that was different depending on the dose and time of exposure. In agreement with these changes, ZEN altered circulating adipokines concentrations, inducing significant changes in adiponectin, resistin, and fetuin B. Our results suggest that ZEN may function as a natural metabolism-disrupting chemical.

## 1. Introduction

The term endocrine disruptors (EDs) refers to those natural or synthetic chemicals that mimic, block, or interfere with hormones in the body’s endocrine system. Exposure to EDs influences the regulation of body homeostasis and contributes to promote a myriad of health issues [1]. Recent epidemiological and experimental data indicate that exposure to specific EDs, known as metabolism disrupting chemicals (MDCs), can influence metabolism, contributing to the incidence of obesity as well as causing metabolic diseases such as diabetes or fatty liver disease [2]. Most of the studied MDCs are synthetic chemicals such as polychlorinated bisphenols and some pesticides (dichloro-diphenyl-trichloroethane, hexachlorobenzene) [3,4]. Conversely, the role as MDCs of naturally occurring EDs such as phytoestrogens [5] and mycoestrogens [6] remains relatively unexplored.

Zearalenone (ZEN) is a toxic secondary metabolite synthesized by several *Fusarium* species that grow on crops [7]. The structure of the mycoestrogen ZEN resembles the endogenous estrogen 17β-estradiol and is able to bind estrogen receptors [8]. ZEN is, hence, a naturally occurring ED that contaminates food [9] and feed [10]. ZEN is present in a range of food products including cereals, dried fruits, and spices [8]. Due to its toxicity, a tolerable daily intake of 0.25 µg/kg body weight and maximum levels in food for human consumption (20–100 µg/kg) have been set by the European Commission by the Regulation EC No 1126/2007.

The well-known reprotoxic and endocrine disrupting activities of ZEN include infertility, hormonal dysfunctions, and reproductive tract hyperplasia [11]. Conversely, the role of ZEN as a MDC is relatively unexplored, although ZEN possess anabolic properties in farm animals [6,12]. Actually, the ZEN derivative zeranol is employed as an anabolic growth promoter for farm animals in some countries, e.g., for cattle in the US, and belongs to the list of substances prohibited in sports [13]. Exposure to ZEN has an impact on different systemic metabolic processes, including cell membrane metabolism, protein biosynthesis, glycolysis, and gut microbiota metabolism in rats and pigs [14,15,16]. Moreover, the toxicity of ZEN has been related with its ability to disrupt lipid metabolism and lipid-related signaling [14,17]. In spite of this evidence, the influence of ZEN exposure in metabolism-related parameters like adipokines has been scarcely studied. Adipokines are a family of metabolically sensitive protein hormones that signal the functional status of adipose tissue to targets in different tissues [18]. Their main roles are the regulation of inflammatory processes and energy balance, mainly glucose metabolism, as well as cell proliferation and differentiation [19]. Adipokines are also involved in reproductive functions and their dysregulation encompasses fertility disorders [20]. The levels of circulating adipokines reflect the metabolic status of an individual and are altered in the case of exposure to MDCs [21]. Although the impact that ZEN could have in the levels of circulating adipokines has not been studied to date, research data suggest that ZEN could disturb the expression and function of adipokines through the dysregulation of PPARα, PPARγ [17] as well as ERK1/2 and PI3K/Akt [22] signaling pathways.

In this study, we investigated if ZEN exposure in pigs modulates the abundance of different blood parameters related to energy balance, as well as changes in the concentration of an array of adipokines. We discovered that exposure to different doses of ZEN induced varying phenotypes regarding metabolic-related parameters and adipokine profiles, and that such changes had different kinetic effects. These results suggest that ZEN could induce a metabolic disruption.

## 2. Results

In this study, female piglets were exposed to dietary ZEN concentrations of 680 (ZEN*low* pigs) and 1620 µg/kg (ZEN*high* pigs) for 28 days.

### 2.1. Effect of ZEN on Blood Biochemical Parameters

Different biochemical parameters were quantified in serum from ZEN*low* and ZEN*high* pigs at 7 and 21 days of exposure. The studied parameters included total protein content, markers of liver function as well as several parameters related to glucose and lipid metabolism.

#### 2.1.1. Liver Function Parameters and Total Protein Content

Exposure to ZEN had a limited effect on markers of liver damage (Figure 1). No significant differences in alkaline phosphatase (ALP) nor alanine transaminase (ALT) serum concentrations were observed in any exposure condition. Significant differences in aspartate transaminase (AST) levels were observed at days 7 (*p* = 0.05) and 21 (*p* = 0.044). At day 7, AST levels were higher in the ZEN*low* group than in controls or ZEN*high* group, although Dunn’s post-test revealed no significant differences. At day 21, AST levels were lower in animals exposed to ZEN, with the difference between the ZEN*high* group and the control animals being significant (*p*-value < 0.05). No differences in bilirubin (BIL) concentrations were observed after 7 days exposure to ZEN, whereas both ZEN*low* and ZEN*high* groups showed significantly lower concentrations of BIL after 21 days (*p* = 0.048). Differences in BIL concentration between ZEN*high* and controls were significant according to Dunn’s post-test (*p*-value < 0.05).

No significant differences were observed for total proteins or albumin levels at any exposure condition (Figure 2).

#### 2.1.2. Lipid Metabolism

Exposure to ZEN had a significant effect on biochemical parameters related with lipid metabolism (Figure 3). At day 7, significant differences in low-density (LDL), high-density lipoprotein (HDL), and cholesterol (CHO) levels (*p* = 0.0032, 0.003, and 0.0033, respectively) were observed. Concentrations of all three parameters were higher in ZEN*low* group compared to controls and ZEN*high* pigs, but differences were not significant according to Dunn’s post-test. No significant differences were found in the levels of free fatty acids (FFA) at this time-point. Slightly lower levels of HDL, LDL, and CHO concentrations were observed in pigs exposed to ZEN compared to controls at day 21, although differences were not significant, except for the ZEN*high* group compared to control for HDL (*p*-value < 0.05). In turn, significantly lower levels of FFA were quantified on animals exposed for 21 days to ZEN (*p* < 0.001). Dunn’s post-test was significant for FFA concentrations in ZEN*low* and ZENh*igh* groups compared to control (*p* < 0.01 and *p* < 0.001, respectively).

#### 2.1.3. Glucose Metabolism

Exposure to ZEN had a significant effect on biochemical parameters related to glucose metabolism (Figure 4). At day 7, significant differences in glucose (*p* < 0.001) were observed. Compared to control, the glucose concentration was significantly higher in ZEN*low* group compared to controls (*p* < 0.05), whereas it was significantly lower in ZEN*high* pigs (*p* < 0.05). At the same time-point, significantly lower levels of serum lactic acid were observed in ZEN*low* pigs compared to control, whereas no differences were observed in ZEN*high* pigs. At day 21, differences in glucose were also significant (*p* < 0.001). Compared to control, glucose concentrations were significantly lower in both ZEN*low* and ZEN*high* groups (*p* < 0.05). In turn, no differences in lactic acid concentrations were observed at 21 days of exposure to ZEN.

### 2.2. Screening of the Effect of ZEN on the Relative Abundance of Circulating Adipokines

In order to obtain an overview of the changes induced by exposure to ZEN in circulating adipokines, pools of serum were prepared from samples collected from ZEN*low* pigs at days 7 and 21 and analyzed via adipokines protein arrays. This group was chosen because it showed the most evident changes in metabolic parameters. At day 7, the relative abundance of 32 proteins was more than 1.5-fold less intense in ZEN*low* samples compared to controls, whereas a higher than 1.5 fold-change in relative abundance was observed in 2 proteins. At day 21, 30 protein spots showed a more than 1.5-fold decrease and 11 spots showed a more than 1.5-fold increase in ZEN*low* samples (Table 1). Of the spots showing an apparent down-accumulation at day 7, 18 were also down-accumulated at 21, 8 were up-accumulated at day 21, and 6 did not change at day 21. Of the two spots showing an apparent up-accumulation at day 7, one (M-CSF) was also up-accumulated at day 21, whereas the other was unchanged (resisitin). Regarding spots showing signal changes higher than 1.5 at day 21 but not at day 7, 12 were down-accumulated and 2 were up-accumulated.

### 2.3. Effect of ZEN on the Serum Concentrations of Resistin, Adiponectin, and Fetuin B

Three adipokines, namely resistin, adiponectin, and fetuin B showed abundance changes in the protein array that could potentially explain the deviations in metabolic parameters observed according to their known function. The effect of exposure to ZEN in the circulating levels of these adipokines was analyzed by ELISA in pigs from ZEN*low* and ZEN*high* groups and compared to controls (Figure 5). At day 7, the effect on ZEN exposure was significant for all three proteins (*p*-values *p* = 0.05, 0.025, and 0.046, respectively). Resistin levels were 1.47 and 1.25 times higher than control for both ZENlow and high, respectively, but only significantly higher in ZEN*low* group (*p*-value < 0.05). Adiponectin concentration was significantly lower in ZEN*low* and ZEN*high* groups (1.38 and 1.3 times less abundant, respectively; *p*-value < 0.05). Compared to control, only ZEN*low* pigs showed significantly lower levels of fetuin B, although ZEN*high* pigs also showed a decrease in fetuin B concentrations (1.43 and 1.3-fold decrease in abundance in ZEN*low* and ZEN*high* pigs compared to control).

At day 21, only resistin levels were significantly affected by exposure to ZEN (*p*-value = 0.048). Resistin showed a 1.63-fold decrease in concentration in ZEN*high* pigs compared to controls, whereas was not different for ZEN*low* pigs (1.01 fold-change). Regarding adiponectin, values were lower than controls, but not significant, whereas levels of fetuin B were similar for all groups.

## 3. Discussion

ZEN is a relatively well-known natural ED [8]. Many of the toxic effects of ZEN derive from its binding to estrogen receptors, but other non-classical estrogen triggered pathways can also be activated by ZEN [22], including the ability to change the circulating levels of important metabolic-related hormones such as gonadotropins, insulin, and estrogen [23,24]. The latter effects are known to mediate the metabolic disruption induced by xenoestrogens [25]. However, the role of ZEN as a metabolic disruptor has not been investigated to date. In the present study, we investigated if exposure to different doses of ZEN were related with disturbances in several metabolism-related biochemical serum parameters over time. It is worth mentioning before discussing our results, that the ZEN doses used in the present study are all above the NOEL (10.4 μg/kg body weight). In this sense, the nomenclature of groups in the present study (ZEN*low* and ZEN*high*) is inherent to this study and does not represent realistic low exposure conditions to ZEN. The effects of ZEN-contaminated treatment diets on the body weight and vulva size of piglets included in the present study were already reported in detail in a previous publication [26]. Briefly, ZEN had no influence on the body weight but caused a time- and dose-dependent increase of the vulva area. Compared to the control group, the vulva area was significantly enlarged from day 11 (ZEN*high*) and day 14 (ZEN*low*) onwards. At day 26, the vulva area was increased by factor of 1.8 and 2.9 in the ZEN*low* and ZEN*high* piglets, respectively, compared to the controls.

We first investigated the effect of ZEN in serum parameters related to the liver function. We observed that exposure to ZEN was not associated with hepatocellular damage, since no changes were detected in liver enzymes or total protein content and albumin levels. However, we found minor AST level changes (less than two-fold) depending on ZEN dose and time of exposure. These AST levels suggest minor fatty changes in the liver rather than liver damage since the difference in AST was not accompanied by changes in ALT, and AST/ALT ratios were between 1 and 1.3 in all cases. Moreover, the changes observed in the lipid profile (discussed below) also sustain this hypothesis. A significant drop in bilirubin levels was observed in all exposed animals at 21 days. Serum bilirubin is a major contributor to the total antioxidant capacity in blood plasma, mainly against lipid peroxidation. A decrease in serum bilirubin levels suggests an increase in oxidative stress, and has been linked with a pro-inflammatory state during metabolic syndrome [27].

Exposure to ZEN induced significant changes in the lipid profile. In ZEN*low* animals, an increase of the lipoproteins LDL and HDL as well as cholesterol was observed at day 7, but it was no longer present at day 21. Such changes could be explained either by a rise in intestinal absorption of cholesterol or by an increase in cholesterol synthesis in the liver. Both LDL and HDL levels paralleled the rise in cholesterol, indicating that bidirectional cholesterol transport from liver to peripheral tissues and back was enhanced. Such effects are known for estrogens [28] and xenoestrogens [29], since these regulate liver lipid metabolism. These results, together with AST concentration changes suggest a possible disruption of liver lipid metabolism by ZEN, which should be further investigated.

At day 21, ZEN*low* and ZEN*high* animals showed significantly lower FFA levels. There is a relationship between circulating levels of FFA and glucose metabolism, as high FFA serum levels are often related with the development of insulin resistance [30]. In the present study, animal groups showing lower FFA levels also exhibited lower glucose levels, suggesting a likely insulin-mediated mechanism for both effects. Indeed, changes in glucose metabolism were also observed in animals exposed to ZEN, but they were different depending on the exposure conditions. In ZEN*low* animals, a significant increase in glucose and decrease in lactic acid was observed at day 7, whereas at day 21, glucose levels were significantly decreased and lactic acid levels were not different from controls. In contrast, ZEN*high* animals showed decreased glucose levels and unchanged lactic acid at all measured times. Lactic acid and glucose are linked through both glycolysis and gluconeogenesis, as gluconeogenesis recycles circulating lactic acid into glucose. Our results suggest that a short exposure to a lower dose of ZEN increases serum glucose levels, whereas a longer exposure to a lower dose or exposure to a higher dose decrease glucose and FFA levels. This apparently contradictory effect has been identified before in pigs exposed to much lower ZEN concentrations [16], and can be explained by the dual effect of estrogens and xenoestrogens in glucose metabolism. Estrogen suppresses liver gluconeogenesis [27] but xenoestrogens such as bisphenol S can increase it [28]. In the pancreas, estrogen regulates the biosynthesis and release of insulin. Activation of ERalpha by 17beta-estradiol and the environmental estrogen bisphenol-A (BPA) promotes an increase of insulin biosynthesis in pancreas beta-cells [31]. This modulation of insulin synthesis by (xeno)estrogens can have positive or negative effects. Indeed, estrogens promote the adaptation of the endocrine pancreas to pregnancy, and the administration of xenoestrogens can reverse the diabetic effects of the insulin-deficiency inducer streptozotocin [31]. However, xenoestrogens can also provoke an excessive insulin signaling, contributing to the development of type II diabetes [32]. In any case, the influence of ZEN in glucose metabolism in pigs observed in the present study seems especially relevant, because commercial pigs are resistant to the development of type II diabetes, even if they are bred following a diabetogenic lifestyle [33]. Indeed, the current results warrant further research on the role of ZEN in insulin signaling and glucose tolerance.

The changes in metabolism such as the ones described above often involve changes in the signaling of the protein hormones adipokines, which are also known to have important functions in the reproductive system. Thus, changes in circulating adipokines might help relating to the disruption of metabolic and reproductive functions induced by ZEN. We aimed at obtaining an overview of the changes in circulating adipokine concentrations, and because animals from the ZEN*low* group showed the most pronounced changes in biochemical parameters, we used a protein array with sera pools of these pigs. Results indicate that ZEN has a time-dependent effect on the abundance of multiple mediators of inflammation, coagulation, cell growth, and metabolism.

A few parameters including several growth promoters (GH, TGF-B1) appeared depleted from serum at day 7 and were found at a normal ratio at day 21, whereas other molecules remained down-accumulated also at day 21, including growth factors (e.g., BMP-4, FGF-19), cytokines (e.g., IL-10, TNFα), and the adipokine hormones visfatin, leptin, and chemerin. These three adipokines are important because they are energy sensors, their expression is influenced by estrogens, and they act in the interphase between metabolism and reproduction. The adipokine visfatin has hypoglycemic effects and is involved in the regulation of female fertility. The expression of visfatin is upregulated by estrogen, inflammation, and hyperglycemia and downregulated by insulin [34]. However, as opposed to estrogens and in agreement with our results, the exposure to the xenoestrogen genistein is associated with a decrease in the circulating levels of visfatin in humans [35]. Leptin reduces food intake and increases energy expenditure [36]. In swine, there is a positive correlation between circulating leptin in plasma and adipose tissue mass, and this correlation regulates the reproductive neuroendocrine axis in gilts [37]. Leptin expression is under nutritional and hormonal regulation, and sex hormones are important regulators of leptin synthesis [36]. Our results indicate a depletion of circulating leptin upon exposure to ZEN, in agreement with the effect of other xenoestrogens, such as genistein and parabens, that are known to inhibit its synthesis [38,39]. However, other studies indicate that BPA induces leptin synthesis in humans, whereas the ZEN derivative zeranol does not influence leptin levels in cattle [40]. These results suggest that the exact mechanism of action of each molecule, as well as exposure conditions, could result in a different modulation of leptin concentrations, which should be taken into account to further investigate the influence of ZEN on leptin levels. Chemerin is a hormone classified as both a cytokine and an adipokine. Chemerin regulates energy homeostasis, fat and glucose metabolism, and modulates insulin sensitivity [41]. In pigs, chemerin is a modulator of ovarian steroidogenesis, and serum chemerin levels are negatively correlated with the onset of puberty [37]. The apparent depletion of circulating chemerin from animals exposed to ZEN observed here is in agreement with the downregulation of the ovarian expression of chemerin mediated by the xenoestrogen BPA [42]. Future studies should contemplate monitoring the circulating levels of these three molecules following exposure to ZEN in order to investigate their connection with metabolic and reproductive disruption.

Eight molecules presented an apparent serum depletion at day 7 and an accumulation at day 21. These molecules were the estrogen-regulated proteins fibroblast growth factor-basic (FGF-2), angiotensinogen and entactin, the pro-inflammatory molecules interleukin 1 beta (IL-1β), interleukin 6 (IL-6) and myeloperoxidase (MPO), the hepatokine fetuin B, and the adipokine hormone adiponectin. An adipokine hormone (vaspin) and an acute phase protein (C-reactive protein) were not altered at day 7 but accumulated at day 21, whereas the adipokine hormone resistin was accumulated at day 7 and unchanged at day 21. Two molecules involved in inflammation, Colony Stimulating Factor 1 (M-CSF) and C-C Motif Chemokine Ligand 5 (RANTES) were accumulated at both time-points. These changes indicate the development of a pro-inflammatory state around 21 days of exposure, which is consistent with the decrease in bilirubin discussed above. Estrogens are known to have an immunomodulating role thus having anti-inflammatory but also pro-inflammatory roles depending on a number of factors including differential estrogen receptor activation and hormone concentration [43]. Many xenoestrogens mediate obesity-associated inflammation and related metabolic abnormalities. Our results are in agreement with previous reports [44,45] indicating that ZEN might be responsible for the development of a pro-inflammatory and pro-oxidant status, although further studies should confirm the role of ZEN in the context of metabolic inflammation.

The relative abundance changes observed for resistin, adiponectin, and fetuin B in the protein array could explain some of the changes on metabolic parameters described above, mainly changes in glycemia. The absolute quantification of the circulating levels of these three proteins using ELISA assays confirmed the trends in protein changes detected using the protein array.

According to the ELISA results, resistin levels were more abundant in ZEN*low* animals than controls at 7 and 21 days of exposure, while concentrations did not show differences in ZEN*high* animals. In contrast, the concentration of adiponectin was reduced at both ZEN*low* and ZEN*high* animals but only at day 7. The increase of the circulating levels of resistin and the decrease of adiponectin observed upon exposure to ZEN is accompanied by changes in glucose levels. Resistin and adiponectin are antagonist molecules, since resistin induces insulin resistance and is considered pro-inflammatory, whereas adiponectin induces insulin sensitivity and is considered protective against inflammation [46]. In the pig, resistin is suspected to be involved in fattening, while adiponectin levels are negatively correlated with adiposity [37]. Both molecules have important roles in regulating sexual maturity [37] and their release is affected by sexual hormones including estrogen. The results observed in our study are in agreement with the effect of other xenoestrogens such as octylphenol [47] and BPA [48], and suggest that ZEN is able to disrupt glucose metabolism at least transiently in a mechanism that might involve the regulation of circulating resistin and adiponectin abundance.

The quantification of circulating fetuin B levels indicated a significant reduction only in ZEN*low* animals at day 7. Fetuin B is a hepatokine known to cause glucose intolerance, and its reduction is associated with an improvement of the body’s ability to dispose of a glucose load [49]. Fetuin B is also known to have a role in reproduction, as fetuin-B deficiency renders female mice infertile [50,51]. According to our results, the reduction of fetuin B levels coincide with an increase in glucose concentration, suggesting that fetuin B decrease might be a response to increased glycemia. The regulation of fetuin B levels by xenoestrogens is mostly unknown but given the apparent important impact of fetuin B deletion in female reproduction, it would be essential to understand if changes in circulating fetuin B levels might be directly or indirectly regulated by ZEN.

## 4. Conclusions

The relationship between the exposure to some endocrine disruptors and the development of obesity and metabolic disease has been stated only recently. Obesity is a complex endocrine disease that is caused by the disruption of many hormonal control systems, and it is related with the development of the condition known as metabolic syndrome, which includes a myriad of symptoms such as insulin resistance, hyperglycemia, and dyslipidemia. Although several synthetic and natural xenoestrogens are known to be metabolic disruptors, little is known about ZEN. Our results showed that ZEN is able to prompt changes in lipid metabolism without inducing liver damage. However, some of the observed changes indicate that alterations in lipid metabolism could have effects in the liver, supporting further research on the role of ZEN exposure in the context of fatty liver disease. ZEN exposure also seemed to promote a pro-inflammatory status, as well as changes in glucose metabolism that suggest an influence in liver glucose metabolism and insulin signaling. According to our results, these changes might be facilitated by a ZEN-mediated modulation of several adipokines known to have a pivotal role between metabolism and reproduction. In all, the present study shows that ZEN induces alterations in metabolism and circulating adipokines, suggesting that this mycotoxin might act as a metabolic disruptor. There is a possibility that the described changes and the reproductive toxicity induced by ZEN might be interrelated, which merits further research.

## 5. Materials and Methods

### 5.1. Animal Experiment

All procedures for animal handling, care and treatment of pigs have been approved by the institutional ethics committee of the Vetmeduni Vienna and the national authority according to paragraph 26 of Law for Animal Experiments, Tierversuchsgesetz 2012-TVG 2012 (BMBWF-68.205/0058-V/3b/2018).

The experiment was carried out at facilities of the University Clinic for Swine, Vetmeduni Vienna. Weaned, four-week-old, female crossbred piglets (sow: Large White, boar: Pietrain) were obtained from a university-owned pig farm in Lower Austria. At arrival, piglets were allocated to different groups (*n* = 10) considering a balanced average body weight among groups (7.40 kg). Piglets were housed in pens (1 pen/group) on straw bedding and had free access to water and feed during the whole trial period.

After an acclimatization period of 8 days in which all piglets received uncontaminated basal feed, animals were exposed to different treatment diets for 28 days. Piglets received either uncontaminated basal feed (control) or feed with a target contamination of 500 (ZEN*low*) and 1500 µg/kg ZEN (ZEN*high*). For artificial contamination of diets, culture material of Fusarium graminearum was used (BiMM–Bioactive Microbial Metabolites Group, Universitäts- und Forschungszentrum, Tulln, Austria). Details regarding the mixing procedure, determination of final dietary ZEN levels of 680 (ZEN*low*) and 1620 µg/kg (ZEN*high*), as well as absence of relevant co-contamination of diets with other regulated mycotoxins can be retrieved elsewhere [52].

During the experimental period, the general condition of the piglets was checked daily. On days 7 and 21, blood was collected from individual piglets. After centrifugation (3756 rcf, 10 min, 20 °C), serum samples were stored at −80 °C until further analysis.

### 5.2. Clinical Biochemistry Analysis

Serum biochemistry parameters were determined with a Pentra 400 Clinical Chemistry benchtop analyzer (Horiba, Les Ulis, France) at GenoToul-Anexplo platform (Toulouse, France). Serum L(+)-lactate concentration was determined following the manufacturer’s instructions of Lactate Assay Kit II (Sigma, St. Quentin Fallavier, France).

### 5.3. Adipokines Protein Array

The Proteome Profiler Human Adipokine Array Kit (R&D Systems, Minneapolis, MN, USA) was used in order to obtain an overview of the changes induced by exposure to ZEN in circulating adipokines. According to changes in serum biochemistry parameters, changes from ZEN*low* animals were more important than that of ZEN*high*, and so five serum samples from ZEN*low* groups as well as controls at 7 and 21 days of exposure were randomly selected and pooled. These pools were incubated with a cocktail of biotinylated detection antibodies, and then incubated with the nitrocellulose membranes containing spotted capture antibodies. The immobilized adipokine–antibody complexes were then detected using Streptavidin-Atto-680 and fluorescent detection. Images were obtained using a Li-Cor Odyssey Infrared Imager (Li-Cor Biosciences, Lincoln, NE, USA) and analyzed with Image Studio Lite Software v5.2 (Li-Cor Biosciences, Lincoln, NE, USA). Protein abundance was relative to spot signal. Results were expressed as relative fold-change in signal intensity between exposed and control groups. A fold-change in abundance of 1.5 between controls and exposed animals was considered relevant.

### 5.4. Quantification of Resistin, Adiponectin, and Fetuin B by ELISA

The concentration of resistin, adiponectin, and fetuin B were determined using the commercial enzyme-linked immunoenzymatic assays in all serum samples. The ELISA kits employed were the Human Resistin Quantikine ELISA Kit (R&D Systems, Minneapolis, MN, USA), the Pig Adiponectin (ADIPOQ) ELISA Kit (Abbexa Ltd., Cambridge, UK), and the Pig Fetuin B (FETUB) ELISA Kit (Abbexa Ltd., Cambridge, UK).

### 5.5. Statistical Analysis

As data did not pass D’Agostino and Pearson omnibus normality test, non-parametric statistical analysis was applied. Statistical differences were determined using Kruskal–Wallis and Dunn’s multiple comparisons tests using GraphPad Prism statistical software version 6 (GraphPad Software, San Diego, CA, USA). The significance level was set at *p* < 0.05.

## Figures and Tables

**Figure 1 toxins-13-00790-f001:**
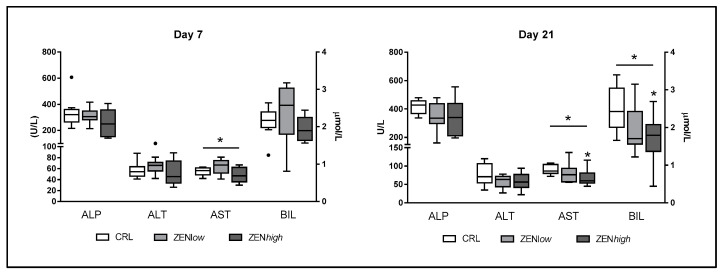
Concentrations of biochemical parameters related to liver status in the serum of pigs exposed to low concentrations of ZEN (ZEN*low*; *n* = 10), high doses of ZEN (ZEN*high*; *n* = 10), and control (*n* = 10). Alkaline phosphatase (ALP), alanine transaminase (ALT), and aspartate transaminase (AST) values are expressed in units per liter (U/L) and plotted on the left axis, whereas bilirubin values (BIL) are expressed in micromoles per liter (µmoles/L) and plotted on the right axis. The plot illustrates the median (line within box); 25th and 75th percentiles (box); 1.5 times the inter-quartile distance (Tukey; whiskers); and outliers (•). Asterisks over a bar indicate the level of significance of Kruskal–Wallis test, whereas an asterisk over the whisker of a group indicates a significant difference of that group with the control group according to Dunn’s post-test. * *p* < 0.05.

**Figure 2 toxins-13-00790-f002:**
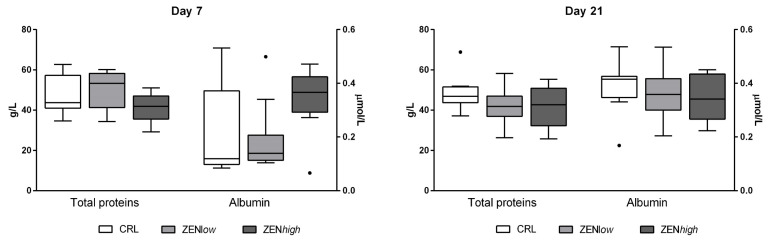
Concentrations of total proteins and albumin in serum of pigs exposed to low concentrations of ZEN (ZEN*low*; *n* = 10), high doses of ZEN (ZEN*high*; *n* = 10), and control (*n* = 10). Total proteins are expressed in grams per liter (g/L) and plotted on the left axis, whereas albumin values are expressed in micromoles per liter (µmoles/L) and plotted on the right axis. The plot illustrates the median (line within box); 25th and 75th percentiles (box); 1.5 times the inter-quartile distance (Tukey; whiskers); and outliers (•). Asterisks over a bar indicate the level of significance of Kruskal–Wallis test, whereas an asterisk over the whisker of a group indicates a significant difference of that group with the control group according to Dunn’s post-test.

**Figure 3 toxins-13-00790-f003:**
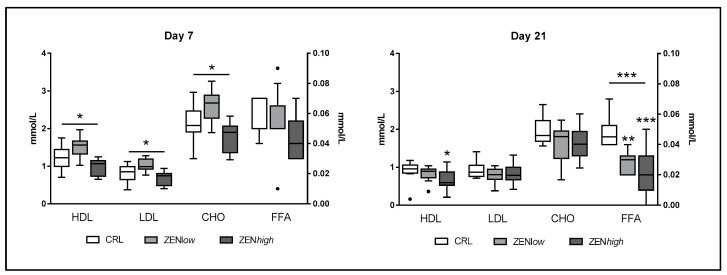
Concentrations of biochemical parameters of lipid metabolism in the serum of pigs exposed to low concentrations of ZEN (ZEN*low*; *n* = 10), high doses of ZEN (ZEN*high*; *n* = 10), and control (*n* = 10). All parameters are expressed in millimoles per liter (mmol/L). High-density lipoprotein (HDL), low-density lipoprotein (LDL), and cholesterol (CHO) values are plotted on the left axis, whereas free fatty acids (FFA) values are plotted on the right axis. The plot illustrates the median (line within box); 25th and 75th percentiles (box); 1.5 times the inter-quartile distance (Tukey; whiskers); and outliers (•). Asterisks over a bar indicate the level of significance of Kruskal–Wallis test, whereas an asterisk over the whisker of a group indicates a significant difference of that group with the control group according to Dunn’s post-test. *** *p* < 0.001; ** *p* < 0.01; * *p* < 0.05.

**Figure 4 toxins-13-00790-f004:**
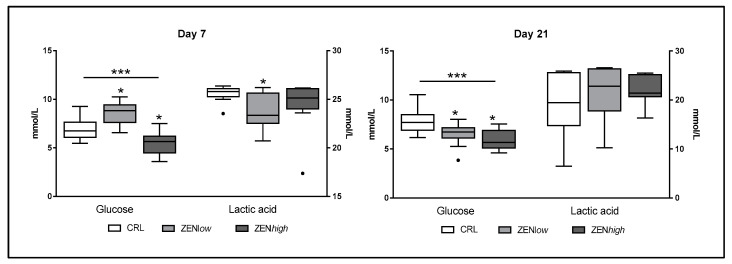
Concentrations of biochemical parameters of glucose metabolism in the serum of pigs exposed to low concentrations of ZEN (ZEN*low*; *n* = 10), high doses of ZEN (ZEN*high*; *n* = 10), and control (*n* = 10). All parameters are expressed in millimoles per liter (mmol/L). Glucose values are plotted on the left axis, whereas acid lactic values are plotted on the right axis. The plot illustrates the median (line within box); 25th and 75th percentiles (box); 1.5 times the inter-quartile distance (Tukey; whiskers); and outliers (•). Asterisks over a bar indicate the level of significance of Kruskal–Wallis test, whereas an asterisk over the whisker of a group indicates a significant difference of that group with control group according to Dunn’s post-test. *** *p* < 0.001; * *p* < 0.05.

**Figure 5 toxins-13-00790-f005:**
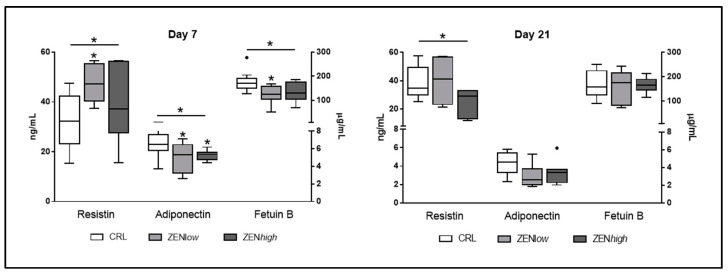
Concentrations of circulating adipokines resistin, adiponectin, and fetuin B in serum of pigs exposed to low concentrations of ZEN (ZEN*low*; *n* = 10), high doses of ZEN (ZEN*high*; *n* = 10), and control (*n* = 10). The concentration of resistin is expressed in nanograms per milliliter (ng/mL) and plotted on the left axis. Adiponectin and fetuin B concentrations are expressed in micrograms per milliliter (µg/mL) and plotted on the right axis. The plot illustrates the median (line within box); 25th and 75th percentiles (box); 1.5 times the inter-quartile distance (Tukey; whiskers); and outliers (•). Asterisks over a bar indicate the level of significance of Kruskal–Wallis test, whereas an asterisk over the whisker of a group indicates a significant difference of that group with the control group according to Dunn’s post-test. * *p* < 0.05.

**Table 1 toxins-13-00790-t001:** Protein name, description and fold-change (FC) values of their abundance in ZEN*low* pig sera (Z) at 7 and 21 days of exposure compared to controls (C). The list is organized by groups of proteins having similar functions.

Protein Name	Symbol	Description	FC Day 7 Z/C	FC Day 21 Z/C
Adiponectins				
Adiponectin	ADIPOQ	Adipokine with wide-ranging paracrine and endocrine effects on metabolism and inflammation.	0.6	1.65
Chemerin	RARRES2	Adipokine that regulates adipogenesis, metabolism, and inflammation through activation of the chemokine-like receptor 1. Implicated in the coagulation, fibrinolytic, and inflammatory cascades	0.71	0.73
Fetuin B	FETUB	Cystatin that promotes phagocytosis of apoptotic cells and plays a role in bone development, calcium homeostasis, and insulin sensitivity. Regulates insulin and hepatocyte growth factor receptors, as well as response to systemic inflammation.	0.48	1.6
Leptin	LEP	Protein secreted by white adipocytes into the circulation. Plays a major role in the regulation of energy balance and body weight control.	0.55	<0.01
Resistin	RETN	Hormone that suppress insulin ability to stimulate glucose uptake into adipose cells. Potentially links obesity to diabetes.	1.55	0.81
Serpin A12	SERPINA12	Adipokine that modulates insulin action in white adipose tissues.	0.67	11.42
Visfatin	NAMPT	Member of the nicotinic acid phosphoribosyltransferase (NAPRTase) family. Involved in metabolism, stress response, and aging. Behaves both as a cytokine with immunomodulating properties and an adipokine with anti-diabetic properties.	0.6	0.63
Inflammation mediators				
Colony Stimulating Factor 1 (Macrophage)	M-CSF	Cytokine that controls the production, differentiation, and function of macrophages.	1.6	2.07
Macrophage Migration Inhibitory Factor	MIF	Lymphokine involved in cell-mediated immunity, immunoregulation, and inflammation.	1.29	0.8
Tumor Necrosis Factor alpha	TNFα	Multifunctional proinflammatory cytokine	0.58	0.13
Tumor Necrosis Factor Superfamily Member 13b	BAFF/BLyS/TNFSF13B	Cytokine playing a role in the stimulation of B- and T-cell function and the regulation of humoral immunity.	0.82	0.22
Interleukin 1 Beta	IL-1B	Potent pro-inflammatory cytokine. Induces prostaglandin synthesis, neutrophil influx and activation, T-cell activation and cytokine production, B-cell activation and antibody production, and fibroblast proliferation and collagen production. Plays a role in angiogenesis.	0.24	1.6
Interleukin 10	IL-10	Major immune regulatory cytokine with profound anti-inflammatory functions.	0.09	0.07
Interleukin 11	IL-11	Cytokine that stimulates the proliferation of hematopoietic stem cells and megakaryocyte progenitor cells and induces megakaryocyte maturation resulting in increased platelet production. Additionally promotes the proliferation of hepatocytes in response to liver damage.	0.46	1.27
Interleukin 6	IL-6	Cytokine with a wide variety of biological functions in immunity, tissue regeneration, and metabolism. Potent inducer of the acute phase response.	0.32	1.6
C-C Motif Chemokine Ligand 2	CCL2/MCP-1	Chemokine that binds the receptor CCR2 and induces the chemoattraction of mononuclear cells. Induces the activation of monocytes, NK cells, lymphocytes, and basophils.	0.95	0.63
C-C Motif Chemokine Ligand 5	CCL5/RANTES	Chemokine with a primary role in the inflammatory immune response via its ability to chemoattract leukocytes and modulate their function.	1.44	1.27
C-X-C Motif Chemokine Ligand 8	CXCL8/IL-8	CXC chemokine with pro-inflammatory effects, involved in angiogenesis.	0.23	0.07
Leukemia Inhibitory Factor Interleukin 6 Family Cytokine	LIF	Pleiotropic cytokine. Induces the hematopoietic differentiation in normal and myeloid leukemia cells, the induction of neuronal cell differentiation, and the stimulation of acute-phase protein synthesis in hepatocytes.	0.57	0.38
Oncostatin M	OSM	Member of the leukemia inhibitory factor/oncostatin-M family of proteins. Regulates cytokine production.	0.29	0.14
Inflammation-related proteins				
C-Reactive Protein	CRP	Member of the pentraxin family of proteins. Sensor and activator of the innate immune response. The level of this protein in plasma increases greatly during acute phase response to tissue injury, infection, or other inflammatory stimuli.	1.05	1.51
S100 Calcium Binding Protein A12	EN-RAGE/S100A12	Calcium-, zinc-, and copper-binding protein with a role in the regulation of inflammatory processes and immune response.	0.98	0.63
Lipocalin-2	LCN2	Iron-trafficking protein involved in multiple processes such as apoptosis, innate immunity, and renal development. Transports small hydrophobic molecules such as lipids, steroid hormones, and retinoids.	0.68	0.78
Myeloperoxidase	MPO	Heme protein, part of the host defense system of polymorphonuclear leukocytes.	0.25	1.6
Pentaxtrin-3	PTX3	Member of the pentraxin family of proteins. Acts as a pattern recognition receptor with roles in the innate immune response to several microbes. Overexpression leads to enhanced pro-inflammatory responses.	0.65	0.98
Cathepsins				
Cathepsin D	CTSD	Enzyme with pepsin-like activity. Plays a role in protein turnover and in the proteolytic activation of hormones and growth factors.	0.66	0.48
Cathepsin L	CTSL	Lysosomal cysteine protease. Hydrolyzes a number of proteins, including the proform of urokinase-type plasminogen activator.	0.46	0.75
Cathepsin S	CTSS	Lysosomal cysteine protease of the papain family. Key protease responsible for the removal of the invariant chain from major histocompatibility complex (MHC) class II molecules and MHC class II antigen presentation.	0.84	1.07
Angiopoietins				
Angiopoietin-1	ANGPT1	Binds the Tie-2 receptor tyrosine kinase. Important modulator of angiogenesis.	0.73	0.79
Angiopoietin-2	ANGPT2	Member of the angiopoietin family. Upregulated in multiple inflammatory diseases. Implicated in the direct control of inflammation-related signaling pathways.	0.68	0.51
Angiopoietin-like 2	ANGPTL2	Binds the Tie-2 receptor tyrosine kinase. Important modulators of angiogenesis.	0.69	0.36
Angiopoietin-like 3	ANGPTL3	Binds the Tie-2 receptor tyrosine kinase. Important modulator of angiogenesis.	0.75	<0.01
Metalloproteinases-related proteins				
Pappalysin-1	PAPPA	Secreted metalloproteinase, which cleaves insulin-like growth factor binding proteins, resulting in activation of the insulin-like growth factor pathway. The encoded protein plays a role in bone formation, inflammation, and wound healing.	0.46	1.42
Tissue Inhibitor Of Metalloproteinases 1	TIMP-1	Metalloproteinase inhibitor. Irreversibly inactivates metalloproteinases by binding to their catalytic zinc cofactor.	0.21	0.21
Tissue Inhibitor Of Metalloproteinases 3	TIMP-3	Inhibitors of the matrix metalloproteinases. Irreversibly inactivates metalloproteinases by binding to their catalytic zinc cofactor. May form part of a tissue-specific acute response to remodeling stimuli.	0.54	0.4
Serine-related proteins				
Complement Factor D	CFD	Serine protease that catalyzes the initial proteolytic step in the alternative pathway of complement.	0.26	<0.01
Serpin A8/Angiotensinogen	AGT	Protease inhibitor involved in vasoconstriction and aldosterone release.	0.47	1.6
Serpin E1	SERPINE1	Serine protease inhibitor. Required for fibrinolysis downregulation and responsible for the controlled degradation of blood clots.	<0.01	<0.01
Dipeptidyl Peptidase 4	DPP4/DPPIV/CD26	Serine exopeptidase that regulates multiple aspects of immune and endocrine function.	0.77	0.49
Vascular-related functions				
Fibrinogen	FG	Major function in hemostasis as one of the primary components of blood clots.	0.02	0.12
Advanced Glycosylation End-Product Specific Receptor	RAGE	Multiligand receptor that mediates both acute and chronic vascular inflammation in conditions such as atherosclerosis and in particular as a complication of diabetes.	0.97	0.73
Vascular Endothelial Growth Factor A	VEGF	Growth factor active in angiogenesis, vasculogenesis, and endothelial cell growth.	0.62	0.25
Endocan	ESM1	Dermatan sulfate proteoglycan involved in angiogenesis and regulated by cytokines.	1.08	0.49
Growth hormones				
FGF basic	FGF2	Growth factor that functions in angiogenesis, wound healing, tissue repair, learning and memory, and the morphogenesis of heart, bone, and brain. It is upregulated in response to inflammatory stimuli and in many tumors.	0.6	8.3
Fibroblast Growth Factor 19	FGF-19	Heparin-binding protein, suppresses bile acid synthesis, and enhances hepatic protein and glycogen synthesis.	0.22	0.12
Growth Hormone	GH	Member of the somatotropin/prolactin family of hormones which play an important role in growth control.	0.54	0.71
Hepatocyte Growth Factor	HGF	Binds to the hepatocyte growth factor receptor to regulate cell growth, cell motility, and morphogenesis in numerous cell and tissue types.	0.73	0.61
Intercellular Adhesion Molecule 1	ICAM-I/CD54	Cell surface glycoprotein with roles in cell proliferation, differentiation, motility, trafficking, apoptosis, and tissue architecture.	0.56	0.57

## Data Availability

The data in this study are available in this article.

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
