# Peer review of "Exposure to Zearalenone Leads to Metabolic Disruption and Changes in Circulating Adipokines Concentrations in Pigs"

_toxins, 2021, doi:10.3390/toxins13110790_

Round 1

Reviewer 1 Report

1, Image quality must be improved, especially fig.5

2, Please check if fig.4 in line 184 is incorrectly marked

3, Table 1 shows some adipokines. Authors should classify and analyze them according to their functions. At present, this part is too simple.

4, Whether the contents stated in lines 72-79 are the experimental results obtained by the authors. If yes, please provide relevant figures and statistical results; if not, the paragraph should be included in the discussion rather than the results.

5, Check if the references are cited correctly.

6, Please add some new references.

Author Response

Reviewer

Comments and Suggestions for Authors

>> Authors would like to acknowledge the comments of reviewer 1.

1, Image quality must be improved, especially fig.5

                >> The image quality of all figures has been improved accordingly and uploaded to the submission files

2, Please check if fig.4 in line 184 is incorrectly marked

>> Corrected accordingly

3, Table 1 shows some adipokines. Authors should classify and analyze them according to their functions. At present, this part is too simple.

>> The table has been reorganized accordingly by grouping proteins by their main role or family of proteins.

4, Whether the contents stated in lines 72-79 are the experimental results obtained by the authors. If yes, please provide relevant figures and statistical results; if not, the paragraph should be included in the discussion rather than the results.

>> This section has been moved to discussion accordingly.

5, Check if the references are cited correctly.

>> References have been checked.

6, Please add some new references.

>> New references have been added to the introduction section, accordingly.

Reviewer 2 Report

No easy task to conduct animal testing related to bioactivities of mycotoxins. The study is quite an undertaking, and one that the authors executed with well-designed experiments and comprehensive data analysis. Data presented are informative and clearly demonstrate the potential metabolic disruption of zearalenone on pigs using  adiponectin, resistin, and fetuin B as the biomarkers. 

Author Response

Reviewer 2

No easy task to conduct animal testing related to bioactivities of mycotoxins. The study is quite an undertaking, and one that the authors executed with well-designed experiments and comprehensive data analysis. Data presented are informative and clearly demonstrate the potential metabolic disruption of zearalenone on pigs using  adiponectin, resistin, and fetuin B as the biomarkers.

>> Authors would like to acknowledge the comments of reviewer 2.

Round 2

Reviewer 1 Report

My questions are addressed.